# The Reliability and Validity of Speech-Language Pathologists’ Estimations of Intelligibility in Dysarthria

**DOI:** 10.3390/brainsci12081011

**Published:** 2022-07-30

**Authors:** Micah E. Hirsch, Austin Thompson, Yunjung Kim, Kaitlin L. Lansford

**Affiliations:** School of Communication Science and Disorders, Florida State University, Tallahassee, FL 32306, USA; athompson4@fsu.edu (A.T.); ykim19@fsu.edu (Y.K.); klansford@fsu.edu (K.L.L.)

**Keywords:** dysarthria, speech intelligibility, speech-language pathologists

## Abstract

This study examined the reliability and validity of speech-language pathologists’ (SLP) estimations of speech intelligibility in dysarthria, including a visual analog scale (VAS) method and a percent estimation method commonly used in clinical settings. Speech samples from 20 speakers with dysarthria of varying etiologies were used to collect orthographic transcriptions from naïve listeners $\left(n=70\right)$ and VAS ratings and percent estimations of intelligibility from SLPs $\left(n=21\right)$. Intra- and interrater reliability for the two SLP intelligibility measures were evaluated, and the relationship between these measures was assessed. Finally, linear regression was used to evaluate the relationship between the naïve listeners’ orthographic transcription scores and the two SLP intelligibility measures. The results indicated that the intrarater reliability for both SLP intelligibility measures was strong, and the interrater reliability between the SLP ratings was moderate to excellent. A moderate positive relationship between SLPs’ VAS ratings and percent estimations was also observed. Finally, both SLPs’ percent estimations and VAS ratings were predictive of naïve listeners’ orthographic transcription scores, with SLPs’ percent estimations being the strongest predictor. In conclusion, the average SLP percent estimations and VAS ratings are valid and reliable intelligibility measures. However, the validity and reliability of these measures vary between SLPs.

## 1. Introduction

Broadly, speech intelligibility refers to how well a listener can understand a speaker’s message [1,2]. In clinical settings, intelligibility is often used as an index of communication outcomes for those with speech disorders [1]. For people with dysarthria, reduced speech intelligibility is a near-universal deficit, regardless of the etiology or dysarthria type, and is also used as an index of dysarthria severity [3,4]. Further, improving speech intelligibility is often a target for dysarthria treatment [1]. Therefore, for speech-language pathologists (SLP), measuring intelligibility is vital for assessing and treating individuals with dysarthria.

The American Speech-Language-Hearing Association (ASHA) recommends using standardized and nonstandardized measures to assess dysarthria and its impact on speech intelligibility [5]. Standardized assessments are beneficial tools for SLPs because they can provide objective measurements, which are often needed to justify to insurance providers the SLP treatment services [6]. There are a handful of formal standardized assessments for measuring speech intelligibility, such as the Assessment of Intelligibility in Dysarthric Speech [7] and the Sentence Intelligibility Test [8,9]. These assessments require SLPs to obtain audio recordings of their patients speaking sentence stimuli of varying lengths and then recruit a naïve listener to transcribe the recordings orthographically.

Beyond these formal standardized measures, the orthographic transcription method for measuring intelligibility is commonly used in research settings and is considered the “gold standard” for measuring speech intelligibility [1]. For this method, naïve listeners’ transcriptions are used to derive a percent words correct score, calculated as the number of correctly transcribed words divided by the total number of words and multiplied by 100. However, despite the use of the orthographic transcription method in formal standardized assessments and research settings, SLPs do not often use this method in clinical practice due to time constraints and a lack of access to naïve listeners [9]. Instead, SLPs opt for informal and subjective assessments of speech intelligibility that do not require naïve listeners’ judgments, such as *percent estimations* of intelligibility [10].

### 1.1. Subjective Ratings of Speech Intelligibility

Percent estimation is a subjective measure of intelligibility expressed as the percent of understood speech following a conversation with a speaker with dysarthria [9,10]. Unlike orthographic transcriptions, percent estimations do not require recruiting naïve listeners to rate intelligibility, as the SLP often makes these perceptual judgments alone. This intelligibility measure is commonly used in clinical practice but has received little attention in research applications. Only a handful of studies have examined the use of percent estimations of intelligibility. For example, Yorkston and Beukelman [11] compared several methods of measuring intelligibility, including percent estimations and orthographic transcriptions. Across listeners, they found these two measures were strongly correlated ($r_{\tau }=0.72\;\mathrm{for}\;\mathrm{sentence}\;\mathrm{intelligibility};\;r_{\tau }=0.86\;\mathrm{for}\;\mathrm{word}\;\mathrm{intelligibility}$). However, when considering the individual listeners, intelligibility estimations varied. In another study, Hustad [12] examined the differences between percent estimations and orthographic transcriptions measured from speakers with dysarthria. They found that percent estimations underestimated orthographic transcription scores. Further, the difference between percent estimations and orthographic transcription scores was affected by speaker severity. Specifically, the difference between these two measures was greater for speakers with severe dysarthria. Thus, these results suggest that the reliability and validity of percent estimations vary by the listener- and speaker-related factors.

Another common measure of intelligibility is the visual analog scale (VAS) rating method [13,14,15,16,17]. Like in the percent estimation method, the VAS intelligibility rating method is a subjective measure that requires listeners to estimate the percentage of words they understood from the speaker. The only difference between the VAS and the percent estimation methods is how they are obtained. For the VAS method, listeners estimate intelligibility using a line representing a continuous intelligibility scale to rate the degree of words understood, presented on paper, or on a computer. Their response is then converted into a percentage score. While this method is commonly used in research, it is not commonly used in clinical settings for measuring intelligibility [10]. 

Like percent estimations, VAS ratings are a quick alternative to orthographic transcription scores. However, before adopting VAS ratings or percent estimates as a replacement for orthographic transcriptions, more information is needed regarding the relationship between these measures, as previous findings have been mixed. While previous research has found high correlations between VAS ratings of intelligibility and orthographic transcriptions [16,17,18], the patterns reported in these studies have varied. For example, Adams, Dykstra, Jenkins and Jog [18] found that the average VAS intelligibility ratings overestimated orthographic transcription scores for speakers with Parkinson’s disease and healthy controls. In contrast, both Abur, Enos and Stepp [17] and Stipancic, Tjaden and Wilding [16] reported that VAS intelligibility ratings slightly underestimated orthographic transcription scores. The latter findings are consistent with the results reported for percent estimations of intelligibility in Hustad [12]. 

The mixed conclusions from previous research are likely due to methodological differences. To start, the types of listeners used in studies have varied. For instance, Adams, Dykstra, Jenkins and Jog [18] collected all intelligibility measures from two trained graduate research assistants, while both Abur, Enos and Stepp [17] and Stipancic, Tjaden and Wilding [16] collected data from a larger group of naïve listeners (N=33 and N=50, respectively). Additionally, studies have varied in their design, with some studies utilizing a between-subjects design [16,17] and others using a within-subjects design [12,18]. Notably, many studies investigating the relationship between intelligibility measures focus on the speaker. When focusing on the speaker, these studies often average intelligibility ratings across listeners, yielding a single intelligibility score for each speaker [16,17]. However, a limitation of this method is that it ignores individual listener variability. Therefore, due to these methodological differences, it is unclear how listeners may vary in their intelligibility ratings depending on the type of intelligibility method used. 

Finally, previous literature in this area varied considerably by the dysarthria severity, etiology, or subtype represented. For example, most studies have included speakers with dysarthria secondary to either Parkinson’s disease or multiple sclerosis [16,17,18], while others have included speakers with cerebral palsy [12]. In addition, some of these studies investigated speakers with mild dysarthria and high intelligibility [16], while others investigated speakers with severe dysarthria and low intelligibility [12]. Therefore, there is a need to investigate the relationship between subjective (i.e., VAS ratings and percent estimations) and objective (i.e., orthographic transcriptions) intelligibility measures from speakers with various dysarthria severities, etiologies, and subtypes.

Both VAS ratings and intelligibility estimations are subjective measures. As such, interrater reliability would likely be lower for these measures since individual listeners differ in their criteria for determining the level of intelligibility distortion. Miller [19] describes this as the “internal yardstick” phenomenon, in which listeners differ in their perceptions and judgment criteria for the degree of intelligibility distortion for speakers with dysarthria. Previous research on percent estimations supports this notion, with reports of greater response variability for percent estimates compared to orthographic transcriptions [11,12]. Interestingly, Stipancic, Tjaden and Wilding [16] found that intra- and interrater reliability was greater for VAS intelligibility ratings compared to orthographic transcription scores. Although this finding is surprising, it suggests that VAS intelligibility ratings are at least as reliable as orthographic transcription scores. However, more research is needed to understand the reliability and validity of VAS ratings and percent estimations of intelligibility.

### 1.2. Predicting Naïve Listeners’ Perceptions of Speech

The overall goal of dysarthria management is to improve the patient’s functional communication. Therefore, to maximize the ecological validity of the intelligibility measures used with speakers with dysarthria, these measures should reflect the patient’s communicative environment outside clinical settings (i.e., non-trained, naïve listeners). Formal standardized measures accomplish this by recruiting naïve listeners to provide orthographic transcriptions. However, SLPs are replacing these formal standardized measures with their subjective intelligibility ratings (i.e., percent estimates) [9]. Therefore, it is worth investigating how these subjective SLP measures relate to naïve listeners’ orthographic transcriptions.

It is important to note that listeners familiarized with dysarthric speech perceive speakers with dysarthria to be more intelligible than unfamiliar listeners [20,21,22]. Thus, it is unsurprising that SLPs have been documented to have a perceptual benefit over naïve listeners and perceive speakers with dysarthria to be more intelligible compared to naïve listeners’ perceptions [23,24]. Notably, in one study, this perceptual benefit was only observed for SLPs who worked in medical settings, had more clinical experience, and had higher self-perceived competence working with patients with dysarthria [23]. This information makes it unclear how SLPs’ intelligibility measurements may relate to naïve listeners’ perceptions. Further, it is unclear how the relationship between SLPs’ subjective ratings of intelligibility and naïve listeners’ perception is affected by how intelligibility is measured (i.e., percent estimations vs. VAS ratings).

### 1.3. Current Study 

The current study examined the reliability and validity of two subjective measures of intelligibility made by SLPs, including VAS ratings and the commonly used clinical measure, percent estimations. SLPs often forego formal intelligibility assessments that use naïve listener transcriptions in favor of quicker subjective intelligibility measures [9]. However, before these subjective intelligibility measures can be adopted to replace formal measures that utilize transcriptions collected from naïve listeners (i.e., the SIT or AIDS), we must understand how these SLP measures relate to naïve listeners’ orthographic transcription scores. Finally, both VAS ratings and percent estimations are subjective measures of intelligibility. Therefore, the current study examined how these subjective measures relate to the objective “gold standard” measure of intelligibility, naïve listeners’ orthographic transcriptions. The following research questions were posed: (1) How reliable are subjective SLP intelligibility measures (VAS ratings and percent estimates), as determined by intrarater and interrater agreement? (2) Is there a strong relationship between SLPs’ percent estimates of intelligibility and their VAS intelligibility ratings? (3) Are the subjective SLP intelligibility measures (percent estimations and VAS ratings) predictive of naïve listener intelligibility, as measured by orthographic transcription? The first question examined the intrarater and interrater reliability of these SLP intelligibility measures. The second and third questions examined the convergent validity of these measures, which describes how closely these measures capture the same construct.

We hypothesized that intra- and interrater reliability would be acceptable for the SLP measures. However, consistent with the findings of Yorkston and Beukelman [11], we believed that intrarater reliability would be stronger than interrater reliability due to the “internal yardstick” phenomenon [19]. Furthermore, we hypothesized there would be a strong correlation between the two subjective SLP intelligibility measures (percent estimates and VAS ratings of intelligibility). Finally, we hypothesized that both SLP percent estimates and VAS intelligibility ratings would predict naïve listener intelligibility, as measured by orthographic transcription scores. However, we hypothesized that these subjective SLP intelligibility measures would slightly overestimate naïve listener intelligibility due to the documented perceptual benefit of SLPs [23]. 

## 2. Methods

This study was approved by the Florida State University Institutional Review Board (FSU-IRB STUDY00002322).

### 2.1. Participants

#### 2.1.1. Listeners

**Naïve Listeners:** Seventy naïve listeners were recruited from the crowdsourcing website, Prolific. The naïve listeners’ age ranged from 19 to 74 years old (M=34.8, SD=13.9). Inclusionary criteria for the naïve listeners included (1) having no current speech, language, or hearing disorder(s) per self-report, (2) being a fluent speaker of English, and (3) being located in the United States of America at the time of the study. A total of 28 women, 37 men, 2 agender, 1 genderqueer, and 2 non-binary naïve listeners participated in the study. The majority of naïve listeners were white/Caucasian (n=56). Of the remaining listeners, 6 were Black/African American, 5 were Asian American, 2 were biracial or multiracial, and one participant did not provide racial demographic information. Six listeners were Hispanic/Latino, while 63 were not, and one listener preferred not to provide ethnic demographic information. Although 46 naïve listeners reported being aware of communication disorders, all listeners had no more than incidental experience with communication disorders. Listeners who indicated they were an SLP, SLP assistant, or audiologist (n=1) were excluded from this sample.

**SLP Listeners:** Twenty-one medical SLPs were recruited online via social media advertisements. The SLP listeners’ age ranged from 24 to 67 years old (M=33.8, SD=10.2). Inclusion criteria for the SLP listeners were the same as those for the naïve listeners. However, SLPs also had to hold a certificate of clinical competence in speech-language pathology (CCC-SLP) from the American Speech-Language-Hearing Association (ASHA). Most of the SLPs were women (n=17), 3 were men, and one preferred not to provide their gender identity. Most of the SLPs were white/Caucasian (n=20), and 1 was Asian American. Finally, all SLPs indicated they were not Hispanic or Latino. The SLP listeners’ years of clinical experience ranged from 1 to 41 years, with an average of 8.7 years of experience (SD=9.6). 

#### 2.1.2. Speakers and Speech Stimuli

Previously recorded stimuli from 20 speakers with dysarthria (11 male; 9 female), collected as part of a larger study conducted in the Motor Speech Disorders Lab at Arizona State University, were used in this study. Detailed demographic information of this well-characterized corpus of speakers is provided in previous studies [25,26,27]. Speakers were diagnosed with dysarthria secondary to Parkinson’s disease (n=5), Huntington’s disease (n=5), amyotrophic lateral sclerosis (n=5), or cerebellar ataxia (n=5). The speakers’ dysarthria severity ranged from mild to severe based on expert ratings from two speech-language pathologists. Racial and ethnic demographic information for the speakers is not available.

An adapted version of *The Grandfather* passage [28] was read in a conversational speaking style (the passage stimuli can be found in the supplemental materials for this paper at https://osf.io/sr9aw/ (accessed on 21 June 2022)). Each speaker’s passage reading was recorded phrase-by-phrase, totaling 35 phrases. For the naïve listener ratings, the passage phrases were arranged into roughly two equal blocks. The first block contained the first 18 phrases of the passage and the second block contained the latter 17 phrases. For the SLP listener ratings, the passage phrases were arranged into six blocks. The phrases in each of the six blocks were combined into one audio file for each block. The blocks were created to reduce familiarity effects for the naïve and SLP listener procedures, as described below.

### 2.2. Procedures

The following experimental tasks were completed online and programmed using Qualtrics [29]. The naïve listener and SLP participants were instructed to wear headphones during the experimental procedures. Figure 1 summarizes the experimental procedures for the SLP and naïve listener groups. The following sections explain the procedures in greater detail.

#### 2.2.1. Naïve Listener Procedure

The naïve listeners in this study were randomly assigned to two speakers to rate during the experiment. After obtaining consent and completing a brief demographic questionnaire, listeners completed the transcription task. Before beginning the task, listeners were informed that they would hear phrases spoken by an individual with a speech disorder. They were instructed to listen to each phrase and type out what they heard in the response box. The survey was programmed to allow listeners to play the audio stimuli only once. For this reason, listeners were encouraged to guess if they were uncertain of the spoken word or phrase and to type an “X” into the response box for words they could not understand. Listeners completed this task for two blocks. For the first block, half of *The Grandfather* passage (spoken by the first assigned speaker) was presented phrase-by-phrase. For the second block, the other half of *The Grandfather* passage (spoken by the second assigned speaker) was presented. At the completion of the study, each speaker had transcription data from 4 to 12 naïve listeners (M=7).

#### 2.2.2. SLP Listener Procedure

The SLPs completed intelligibility ratings for all twenty speakers in two randomized blocks. Before collecting the intelligibility measures from the SLPs, they first completed a brief demographic questionnaire which gathered information about their work history, including years of clinical experience and current work setting. Following the questionnaire, SLPs were randomly assigned to one of two blocks, a VAS rating block or a percent estimate block. 

In the VAS block, SLPs were instructed to use a horizontally presented VAS to indicate how much they understood from the sample. The scale was labeled with the anchors of “Cannot Understand Anything” and “Understand Everything” on the left and right, respectively. The responses to each sample were calculated as a percent understood score between 0 and 100. In the percent estimate block, SLPs were asked to “estimate the percent of speech understood” for each speaker by typing a numeric response between 0 to 100 in the response box. 

Within each block, SLPs provided the respective ratings (i.e., percent estimations or VAS ratings) for all 20 speakers. Again, the survey was programmed to allow listeners to play the audio stimuli only once. The order of the speakers and six passage sections were randomized between the two blocks, such that the speaker and passage sections were not repeated between blocks. For example, if the SLP heard the first section of *The Grandfather* passage for speaker AM5 in their first block, then they heard a different section of the passage for AM5 in the second block. Four speakers from each block were randomly presented to the SLPs to rate again to calculate intrarater reliability. 

### 2.3. Measures and Data Preparation

Three intelligibility measures were obtained, including VAS ratings and percent estimations of intelligibility for the SLP listeners and orthographic transcription scores for naïve listeners. For the orthographic transcription scores, the total number of words correctly transcribed was obtained using the *Autoscore* package in R [30]. Then, a percent words correct score was calculated by dividing the total number of correctly transcribed words by the total number of words and multiplying by 100. 

The individual stimulus-level responses for the subjective SLP intelligibility measures were used to examine the reliability of the SLP measures (i.e., Research Question 1) and the correlation between the SLP intelligibility measures (i.e., Research Question 2). To prepare the data for the linear regression analysis (i.e., Research Question 3), the individual intelligibility ratings (i.e., VAS ratings, percent estimations, and orthographic transcription scores) were averaged to obtain mean intelligibility ratings for each speaker. 

### 2.4. Statistical Analysis

Table 1 describes each research question’s statistical approach, interpretation, and evaluation criteria. For the first research question, intrarater reliability for each subjective SLP intelligibility measure (i.e., SLPs’ agreement with their own ratings) was calculated using Pearson correlation coefficient analysis [31]. Next, interrater reliability for each subjective SLP intelligibility measure (i.e., SLPs’ agreement with other SLPs) was calculated using a two-way intraclass correlation coefficient (ICC) analysis. As we were interested in the reliability of ratings from SLPs at an individual level, we focused on the single ICC measure [32]. However, both single ICC and average ICC are reported for completeness. For the second research question, a repeated-measures correlation analysis was completed to examine the strength of the relationship between the two SLP intelligibility measures. Finally, for the third research question, two linear regression models were created to examine whether the SLP VAS ratings (Model 1) and percent estimates (Model 2) were predictive of the naïve listener orthographic transcriptions. 

All data preparation, analyses, and visualizations were completed using the R programming environment [33]. Data cleaning, manipulations, and calculation of descriptive statistics were completed using the *tidyverse* package [34]. The *icc* function of the *irr* package was used to calculate the ICC for the interrater reliability analysis [35]. The *cor.test* function of the *stats* package was used to calculate the Pearson correlation coefficients for the intrarater reliability analysis, and the *lm* function of the *stats* packages was used to run the linear regression models [33]. Linear regression model assumptions were evaluated using the *check_model* function in the *performance* package [36] and the *rcorr* function in the *Hmisc* package [37]. The *rmcorr* function of the *rmcorr* package was used to calculate the repeated measures correlation between the SLP intelligibility estimates and VAS ratings [38]. 

## 3. Results

A cleaned version of the data set, code for the data preparation and analysis, and additional study materials are provided at https://osf.io/sr9aw/ (accessed on 21 June 2022). Table 2 contains the descriptive statistics for the three intelligibility measures averaged across all speakers and within each dysarthria etiology group. Additionally, Figure 2 depicts the distribution of naïve and SLP listener intelligibility measures. Overall, the distribution of responses appears to have a slight negative skew, indicating that both the naïve and SLPs listeners tended to perceive the speakers with dysarthria as relatively intelligible. Across the four dysarthria groups, speakers with dysarthria secondary to Parkinson’s disease were perceived to be the most intelligible, on average. Notably, for the percent estimation measure, SLPs tended to provide estimates in 5- and 10-point increments, despite being able to indicate any value between 1 and 100 (as seen in Figure 2).

### 3.1. Research Question 1

The first research question evaluated the intrarater and interrater reliability of the subjective SLP VAS ratings and percent estimations. Intrarater reliability was examined by randomly selecting four speakers to be rated again by each SLP within each block and calculating Pearson’s correlation coefficients for the two ratings. For the SLPs’ VAS ratings, the correlation between the SLPs’ first and second ratings was strong, $r\left(75\right)=0.87,\;p<0.001$. For the SLPs’ percent estimations, the correlation between the SLPs’ first and second ratings was strong, $r\left(82\right)=0.85,\;p<0.001$. These results suggest that when SLPs rated the same stimuli twice, their two ratings were in agreement for both the VAS and percent estimation methods. 

The interrater reliability was examined using the single and average ICC scores. The single ICC score reflects the interrater agreement for the individual SLP ratings, thus reflecting the listener agreement at the individual SLP level. The average ICC score reflects the interrater agreement for the average SLP intelligibility ratings (i.e., VAS ratings and percent estimations), thus reflecting the collective listener agreement across SLPs. The reliability is interpreted based on the ICC values, where ICC closer to 1.0 indicate better interrater reliability [32]. The two-way single ICC were 0.54 (p<0.001) and 0.56 (p<0.001) for SLP VAS ratings and percent estimations of intelligibility, respectively. The two-way average ICC for SLP VAS ratings and percent estimations was 0.96 (p<0.001) for both intelligibility measures. These results indicate moderate interrater reliability for the *individual* VAS ratings and percent estimates and excellent interrater reliability for the *average* VAS ratings and percent estimates.

### 3.2. Research Question 2

The second research question focused on the relationship between the two SLP intelligibility measures (i.e., VAS ratings and percent estimations). A repeated measures correlation coefficient analysis was conducted to examine this relationship. The correlation between VAS ratings and percent estimations indicated a moderate and positive relationship, $r\left(381\right)=0.66;\;p<0.001$. Figure 3 depicts the individual SLPs’ relationship between their VAS ratings and percent estimations of intelligibility across the 20 speakers (i.e., the lighter lines). Additionally, this figure shows the average relationship between these two measures across all SLPs (i.e., the darker line). 

### 3.3. Research Question 3

The third research question examined the relationship between naïve listeners’ orthographic transcription scores and SLPs’ percent estimations (Model 1) and VAS ratings (Model 2) of intelligibility. Before constructing the linear regression models, the intelligibility measures were averaged for each speaker, and the correlations between naïve listeners’ orthographic transcription scores, both SLPs’ VAS ratings and SLPs’ percent estimations, were examined. The correlations between the naïve listeners’ orthographic transcription scores and the SLPs’ VAS ratings, $r\left(18\right)=0.88,\;p<0.001$, and percent estimations, $r\left(18\right)=0.96,\;p<0.001$, indicated strong positive relationships between these measures. Additionally, the correlation between the average SLP percent estimation and VAS ratings was also strong, $r\left(18\right)=0.90,\;p<0.001$.

Two simple linear regression analyses were created to model the predictive relationship between the naïve listeners’ orthographic transcription scores and SLPs’ percent estimations (Model 1) and VAS ratings (Model 2) of intelligibility. The model results are reported in Table 3. These relationships are also depicted in Figure 4. The results showed that SLPs’ percent estimates (Model 1) and VAS ratings (Model 2) of intelligibility were significant predictors of naïve listeners’ orthographic transcription scores. However, SLPs’ estimations accounted for about 91% of the variance in naïve listeners’ transcription scores, while the SLPs’ VAS ratings accounted for only about 77% of the variance in naïve listeners’ transcription scores. Thus, these results show that while both SLP intelligibility methods are significant predictors of naïve listeners’ transcription scores, the SLPs’ percent estimations are a better predictor of naïve listeners’ intelligibility than the VAS ratings.

## 4. Discussion

In this study, three research questions were posed to evaluate the reliability and validity of two subjective intelligibility measures provided by SLPs, including percent estimations and VAS intelligibility ratings. The results can be summarized as follows: Across all SLPs, intrarater and interrater analyses revealed acceptable reliability for both measures, as indicated by the Pearson correlation coefficient analysis and average ICC score. However, for individual SLPs, the reliability for both intelligibility measures was notably lower, as indicated by the single ICC score. This finding indicates that SLPs intelligibility ratings are more reliable when averaging ratings across SLPs rather than individual ratings. Further, a positive but moderate relationship between SLPs’ VAS ratings and percent estimations was observed, indicating that these two measures moderately describe the same construct. Finally, the results showed that, on average, both SLP intelligibility measures (i.e., VAS ratings and percent estimations) were predictive of the naïve listeners’ orthographic transcription scores. However, the SLPs’ percent estimations explained more of the variance of the naïve listeners’ orthographic transcription scores, indicating it is the better predictor and, therefore, a more valid measure for predicting naive listeners’ orthographic transcriptions.

### 4.1. Percent Estimations vs. VAS Ratings of Intelligibility

The current study simultaneously investigated two subjective measures of speech intelligibility, including the commonly used measure in research, VAS intelligibility ratings, and the commonly used measure in clinical practice, percent estimations. When the reliability of these measures was examined, the two measures performed similarly. That is, both the VAS intelligibility ratings and percent estimates had strong intrarater reliability, indicating that SLPs were consistent and generally agreed with their own VAS ratings and percent estimates when rating a speaker for a second time.

The interrater reliability, or the reliability agreement between SLPs, yielded less promising results. The average ICC score, which reflects the agreement of the average ratings, was excellent for both measures. However, the single ICC score, which reflects the agreement of the individual SLPs’ ratings, was moderate for the two measures. As previously mentioned, the moderate agreement between individual SLPs’ ratings is likely due to the “internal yardstick” phenomenon described above, in which every SLP varies in their evaluation criteria for subjective intelligibility measurements [19]. 

The current finding of greater agreement among multiple SLPs versus single SLP ratings is consistent with previous studies. For example, Abur, Enos and Stepp [17] examined the reliability of VAS intelligibility ratings in two conditions, a minimal exposure condition and an extended exposure condition. In the minimal exposure condition, listeners rated one sentence per speaker in their study, while in the extended exposure condition, listeners rated all sentences for all speakers. The extended exposure condition simulates responses from listeners familiarized with dysarthric speech (i.e., like the SLPs in the current study). Their results showed that, despite the extended exposure, the reliability of familiar listeners’ ratings was improved when a second rater was added. For the less familiar listeners in the minimal exposure condition, more raters were needed to improve the reliability of the VAS intelligibility ratings to a level comparable to the familiar listeners. Like this previous study, the listeners in our study were more reliable when considered as a collective rather than individuals, even though the SLP listeners in the current study were familiar with dysarthria. 

As described above, regarding reliability, SLPs’ VAS intelligibility ratings and percent estimations performed similarly. However, when these two measures were directly correlated to examine their convergent validity, only a positive and moderate correlation was observed ($r_{rm}=0.66$). This finding suggests that while both measures, in theory, quantify speech intelligibility, they are only moderately correlated, suggesting that they may reflect different constructs. This finding highlights the importance of methodological considerations when measuring intelligibility. We hypothesize how these two measures might reflect differing information in the following section.

### 4.2. Predicting Orthographic Transcriptions Using Subjective Measures of Intelligibility

The current investigation was motivated by evidence suggesting that SLPs frequently forgo formal objective intelligibility assessments that utilize orthographic transcriptions in favor of informal subjective measures, such as SLPs’ percent estimations [9,10]. While not frequently used by SLPs to measure intelligibility, we also chose to investigate VAS rating methods, as this method is commonly used within research contexts. To further assess the convergent validity of SLPs’ estimations of intelligibility, we examined their ability to predict naïve listeners’ orthographic transcriptions. Our findings suggest that, on average, SLPs’ percent estimations and VAS intelligibility ratings strongly predict naïve listeners’ orthographic transcriptions. 

The current finding of VAS ratings strongly predicting orthographic transcriptions is in agreement with previous research [16,17]. Interestingly, SLPs’ percent estimates of intelligibility were a stronger predictor of orthographic transcriptions than VAS ratings, based on the R^2^ values from each model. This finding was surprising, given that the two measures are theoretically similar and were highly correlated based on average ratings. However, this discrepancy may be due to different evaluation criteria when providing VAS ratings versus percent estimations. That is, SLPs’ VAS ratings may reflect the speakers’ global dysarthria deficits, while SLPs’ percent estimations capture speech intelligibility. 

For example, in Figure 4, two VAS outliers can be observed (HDM10 and AM1). The SLPs’ VAS ratings for these two speakers, on average, underestimated the orthographic transcription scores from naïve listeners, while the percent estimation scores for these speakers closely aligned with the orthographic transcription scores. Perceptually, these two speakers demonstrated the representative dysarthria characteristics expected for speakers with Huntington’s disease and Ataxia, respectively. HDM10’s speech contained inappropriate silences, prolonged intervals, imprecise consonants, and irregular articulatory breakdowns consistent with hyperkinetic dysarthria [39]. AM1’s speech was characterized by imprecise consonants, excess and equal stress, irregular articulatory breakdowns, and a harsh voice, consistent with ataxic dysarthria [39]. These speakers are similar in that their speech deficits span several speech subsystems, including prosodic, phonatory, and articulatory domains. However, evidence suggests that the articulatory domain is the greatest contributor to decreased speech intelligibility [40,41]. Therefore, SLPs’ VAS intelligibility ratings may reflect the speaker’s global deficits (i.e., all speech subsystems), while percent estimations and orthographic transcriptions reflect deficits of the articulatory subsystem alone. For this reason, VAS intelligibility ratings may better predict dysarthria severity rather than orthographic transcriptions. However, a systematic investigation of the relationship between VAS ratings, dysarthria severity, and orthographic transcriptions is needed to support this hypothesis.

Based on the smoothed regression lines observed in Figure 4, both SLP intelligibility methods tended to underestimate orthographic transcriptions in the lower intelligibility range and overestimate orthographic transcriptions in the upper range. However, this pattern was more pronounced for (1) the lower range of intelligibility compared to the upper and (2) for VAS intelligibility ratings compared to percent estimations. These results are consistent with previous literature. Specifically, both Abur, Enos and Stepp [17] and Stipancic, Tjaden and Wilding [16] observed VAS intelligibility ratings to underestimate orthographic transcription scores in the lower intelligibility range, while Adams, Dykstra, Jenkins and Jog [18] reported VAS intelligibility ratings to overestimate orthographic scores in the mid-to-upper intelligibility range. Similarly, Hustad [12] observed percent estimations to underestimate orthographic transcription measures for the two speakers with severe dysarthria due to cerebral palsy. Taken together, the current findings, in addition to the handful of previous studies, suggest that subjective measures of intelligibility (i.e., VAS ratings and percent estimations) may be less valid at the extreme ends of the intelligibility range when predicting intelligibility measured using orthographic transcriptions. However, more research is needed to validate this finding.

## 5. Clinical Implications

It is crucial to explicitly state what the current findings imply (and do not imply) for clinical practice. First, research question 1 revealed that SLPs demonstrated high intrarater reliability for the two subjective intelligibility measures. In other words, when rating the intelligibility of a speaker twice (using either VAS ratings or percent estimations), the SLPs’ second ratings were consistent and generally agreed with their first ratings. This finding is positive and suggests that, despite these intelligibility measures being subjective, SLPs are internally consistent when using these intelligibility ratings. However, despite the strong internal consistency (i.e., strong intrarater reliability), the *accuracy* of these subjective intelligibility ratings remains unknown. In fact, based on the finding of only moderate reliability between SLPs (i.e., moderate interrater reliability), it is likely that many of the SLPs’ subjective intelligibility measures are far from the speakers’ true intelligibility levels.

Additionally, the findings of research question 2 may have implications for selecting which subjective intelligibility measure to use. That is, the relationship between percent estimations and VAS ratings is only moderate, indicating that these methods are possibly measuring different underlying constructs. In the previous section, we hypothesize how these two measures may be sensitive to different speech subsystems. However, more research is needed to determine the underlying constructs that these methods are measuring. However, these results still have implications for dysarthria management in clinical settings. Specifically, the current findings should caution SLPs from using these subjective measures interchangeably, as they may provide different information. Instead, SLPs should use the same method throughout assessment and treatment for dysarthria. 

Finally, while the findings from research question 3 found SLPs’ percent estimates to be a strong predictor of naïve listeners’ orthographic transcriptions, it is essential to note that these findings do not support the use of percent estimates (or VAS ratings) as a replacement for orthographic transcriptions or formal intelligibility testing. The relationship between SLPs’ ratings (both VAS ratings and percent estimates) and naïve listeners’ orthographic transcriptions was investigated using data *averaged* across SLPs. In other words, the mean rating value across all 21 SLPs was obtained to produce one value per speaker. This process was also applied to the naïve listeners’ transcriptions. Therefore, the current findings only state that the average ratings (i.e., VAS ratings and percent estimates) among 21 SLPs are predictive of the average transcription scores of naïve listeners. Thus, this finding cannot support SLPs’ use of percent estimations, as SLPs often provide the sole estimation and do not rely on a group of SLPs for intelligibility estimations. However, the findings for research question 3 do support the potential use of crowdsourcing perceptual ratings for SLPs to obtain intelligibility measurements. While the use of crowdsourcing tools is not a common practice among SLPs for obtaining perceptual ratings, recent efforts have been made to develop such tools, such as the Communication-Related Parameters in Speech Disorders (KommPaS) [42].

## 6. Limitations and Future Directions

There are some limitations to this specific investigation that should be considered. First, a relatively homogenous sample of SLPs was recruited for this study. Although the years of experience varied between the SLPs, all SLPs in this study worked in a medical setting. Based on results from Borrie, Lansford and Barrett [23], SLPs working in medical settings had higher baseline intelligibility of dysarthric speech compared to SLPs who worked in other settings, such as schools. Therefore, a more diverse sample of SLPs who work in various clinical settings is needed for future research. 

Additionally, because the perceptual tasks for the SLPs and naïve listeners differed, the average ratings per speaker were used to examine the relationship between the subjective SLP intelligibility measures and orthographic transcription (Research Question 3). Thus, our results for this question have limited implications for how well these measures predict orthographic transcriptions at the individual SLP level. However, our interrater reliability analysis revealed that individual SLP ratings only moderately agreed with each other. Thus, there is a need for future research to investigate how well individual SLP ratings predict orthographic transcriptions and what SLP-related characteristics influence the reliability of their intelligibility ratings. 

Another limitation of our study is the stimuli we used. Specifically, while *The Grandfather* passage was used as the speech stimuli for both the SLP and naïve listeners, the stimuli were presented differently to both groups of listeners (i.e., 1/6 of the passage vs. 1/2 of the passage; see Figure 1). These differences may have influenced the results for the orthographic transcription scores, as the naïve listeners heard more of the passage. Additionally, the phrases of *The Grandfather* passage were not controlled for length or lexical complexity, which may have also impacted orthographic transcription scores. Therefore, future studies should control the methods used between listener groups to ensure listeners are rating similar stimuli.

Finally, the current study frames orthographic transcriptions as an *objective* measure and percent estimations and VAS intelligibility ratings as *subjective* measures. This framework has been adopted from previous literature [1]. However, one could argue that orthographic transcriptions maintain a degree of subjectivity. This argument is valid, especially when conceptualizing intelligibility as a speaker-derived measure. Orthographic transcription accuracy for a speaker may vary from listener to listener, indicating subjectivity. Further, orthographic transcriptions may be affected by various listener factors, such as listener familiarity with the speaker or the passage stimuli [1,23]. However, when intelligibility is conceptualized as a listener-derived measure, it is more accurate to describe orthographic transcriptions as an objective measurement because it reflects the percent of words correctly transcribed by the listener. Additionally, to further reduce the subjectivity to orthographic transcriptions, the gold standard usage of this method is with listeners who are naïve to the speaker and the passage stimuli [1]. In conclusion, the objectivity of orthographic transcriptions can be debated. However, this discussion is beyond the scope of the current study. 

## 7. Conclusions

Overall, the results of this study found that, on average, SLPs’ percent estimates and VAS intelligibility ratings are valid and reliable measures for estimating naïve listener intelligibility of dysarthric speech. However, at an individual SLP level, the reliability of percent estimates and VAS intelligibility ratings is moderate, although intrarater reliability was strong. Further, SLPs’ VAS ratings and percent estimates were only moderately correlated, suggesting that these two measures provide unique information despite some overlap. Finally, the average SLP ratings for VAS intelligibility ratings and percent estimates were significant predictors of orthographic transcription scores measured from naïve listeners. Before subjective intelligibility measures can be adopted as evidence-based clinical practices, future work must investigate individual SLP ratings to determine the specific SLP-related factors that contribute to more or less reliable intelligibility estimates. 

## Figures and Tables

**Figure 1 brainsci-12-01011-f001:**
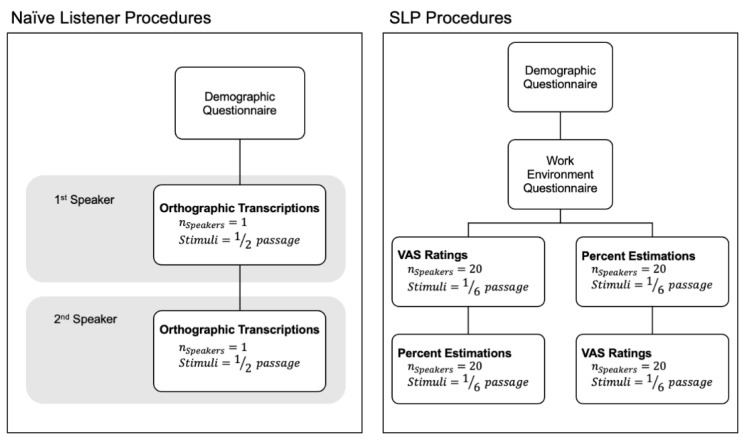
Study Procedures for the SLP and Naïve Listeners. Passage refers to segments of *The Grandfather* passage. Consent was obtained prior to the study procedures depicted in this figure. VAS = visual analog scale ratings.

**Figure 2 brainsci-12-01011-f002:**
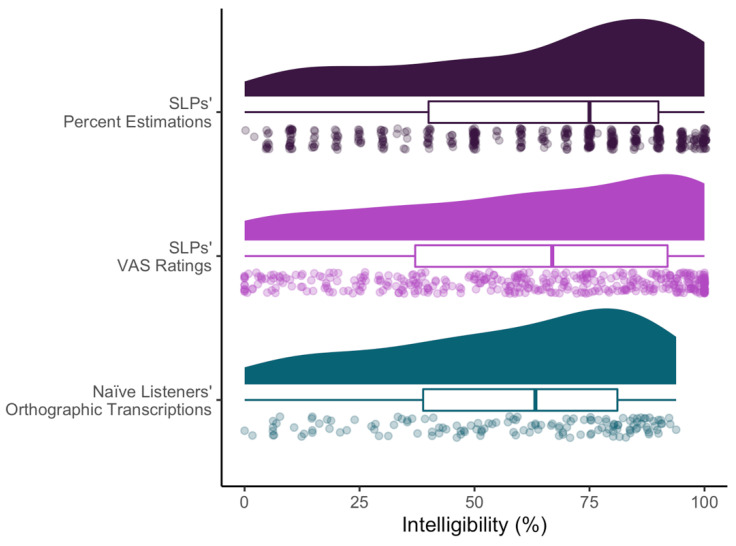
Distribution of Responses for Both the Naïve Listeners’ and SLPs’ Intelligibility Measures. For each of the three ratings investigated in the study, this figure shows the distribution of ratings (top), a box and whisker plot for the ratings (middle), in which the line within the box depicts the median, the left and right ends of the box represent the first and third quartile, respectively, and the left and right “whiskers” depict the smallest and largest rating. Finally, this figure shows the individual listener ratings across all 20 speakers, as indicated by the semi-transparent data points (bottom). Individual data points that appear to be darker signify a high-density region of overlapping responses. SLP = speech-language pathologists, VAS = visual analog scale.

**Figure 3 brainsci-12-01011-f003:**
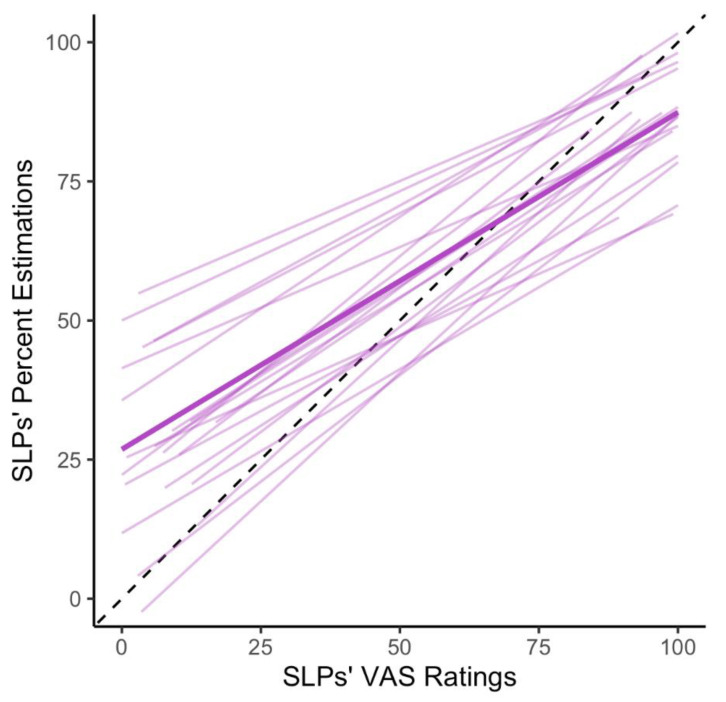
Relationship between SLPs’ VAS Ratings and Percent Estimations. The lighter lines depict the relationship between percent estimations and VAS ratings for each SLP in the study, while the darker line represents the average relationship between these two measures. The black dotted line represents the perfect theoretical relationship between these two measures. Alternate visualization for this figure can be found in the supplementary materials (https://osf.io/sr9aw/) (accessed on 21 June 2022).

**Figure 4 brainsci-12-01011-f004:**
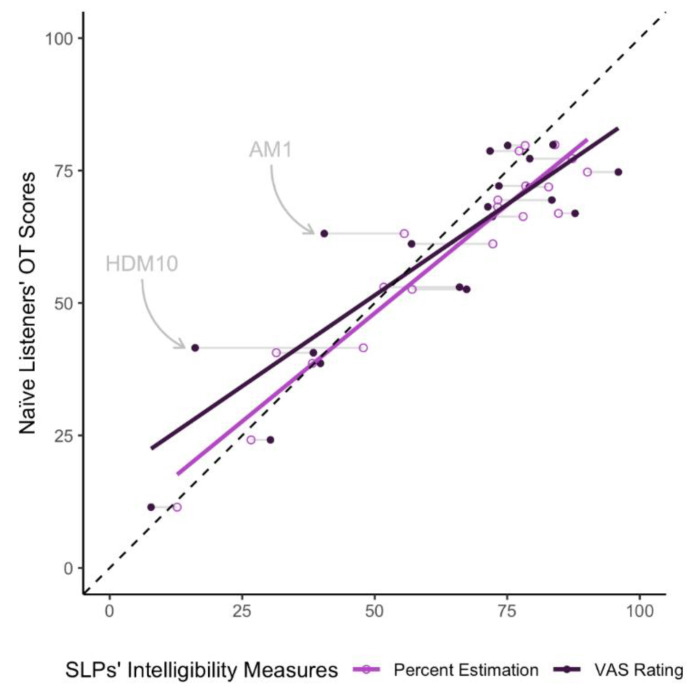
SLPs’ Percent Estimates and VAS Ratings of Intelligibility as Predictors of Naïve Listeners’ Orthographic Transcriptions. OT = orthographic transcription; SLP = Speech-language pathologists, VAS = visual analog scale. The light gray line connects the data points for each of the 20 speakers. Two speakers, AM1 and HDM10, are highlighted within the figure and discussed in Section 4.2 of the discussion.

**Table 1 brainsci-12-01011-t001:** Descriptions of the reliability and validity measures.

Statistical Approach	Interpretation	Evaluation Criteria	
Intrarater Reliability			
*Pearson correlation coefficient (r)* *(Research Question 1)*	SLPs’ agreement with their own intelligibility ratings.	<0.40 0.40–0.75 0.75–0.90 >0.90	Weak Moderate Strong Very Strong
Interrater Reliability			
*Average two-way ICC (ICC)* *(Research Question 1)*	The reliability of the *average* intelligibility ratings across SLPs.	<0.50 0.50–0.75 0.75–0.90 >0.90	Poor Moderate Good Excellent
*Single two-way ICC (ICC)* *(Research Question 1)*	The reliability of *individual* SLP intelligibility ratings.		
Convergent Validity			
*Repeated measures correlation (r_rm_)* *(Research Question 2)*	The strength of the relationship between SLP VAS ratings and percent estimations of intelligibility.	<0.40 0.40–0.75 0.75–0.90 >0.90	Weak Moderate Strong Very Strong
*Linear Regression (R^2^)* *(Research Question 3)*	The predictive relationship between the two SLP intelligibility measures and naïve listener orthographic transcription scores.	The % of the variance of orthographic transcriptions that can be explained by the SLP intelligibility measures.	

Note. ICC = intraclass correlation coefficient. Evaluation criteria for the Pearson correlation coefficient, repeated measures correlation, and regression analyses are provided by Schober, Boer and Schwarte [31]. Evaluation criteria for ICC are provided by Koo and Li [32].

**Table 2 brainsci-12-01011-t002:** Descriptive Statistics for the Three Intelligibility Measures.

	Naïve Listener Measures		SLP Listener Measures			
Etiology	Orthographic Transcription Score		VAS Ratings		Percent Estimations	
	*M*	*SD*	*M*	*SD*	*M*	*SD*
All Speakers	59.56	19.34	61.82	24.66	64.05	22.61
Ataxia	52.04	23.70	50.35	27.71	56.09	26.73
ALS	50.84	19.73	56.35	19.93	54.62	23.31
PD	72.32	5.83	81.09	6.24	80.48	6.74
HD	63.05	20.15	59.47	31.82	65.01	24.40

Note. *M* = mean, *SD* = standard deviation, VAS = Visual Analog Scale.

**Table 3 brainsci-12-01011-t003:** Models Predicting Naïve Listeners’ Orthographic Transcription using SLPs’ Percent Estimations and SLPs’ VAS Ratings.

	Model 1			Model 2		
Predictors	Estimates	CI	p	Estimates	CI	p
(Intercept)	7.23	−1.23–15.69	0.089	17.12	4.70–29.55	**0.010**
SLPs’ Percent Estimations	0.82	0.69–0.94	**<0.001**			
SLPs’ VAS Ratings				0.69	0.50–0.87	**<0.001**
R^2^/R^2^ adjusted	0.913/0.908			0.767/0.754		

Note. Model 1 predicted naïve listeners’ orthographic transcription scores using SLPs’ percent estimations. Model 2 predicted naïve listeners’ orthographic transcription scores using SLPs’ VAS ratings. *p*-values in bold indicate significance at α=0.05.

## Data Availability

The cleaned data, analysis code, and output are provided publicly at https://osf.io/sr9aw/ (accessed on 21 June 2022). This OSF also contains supplementary study materials and additional alternative analyses that were completed but not reported in the current manuscript.

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
