# Peer review of "The Reliability and Validity of Speech-Language Pathologists’ Estimations of Intelligibility in Dysarthria"

_brainsci, 2022, doi:10.3390/brainsci12081011_

Round 1
Reviewer 1 Report
Dear Authors,
Thank you for this useful paper, which helps us to understand more about dysarthria and will be of interest to those SLPs working in this field clinically. Generally this paper was well-explained but there were a few times when I had to read the paper several times to understand. I will try to explain those elements here to improve the reading experience for those interested in your findings.
Abstract - you use a variety of descriptive terms here and I was unclear how these related to each other and what they meant. You used high, moderate, excellent. Is high the same as excellent or in between moderate and excellent? You also used moderate positive. On reading the paper results, these terms are used there too but the variety of terms is confusing without further clarification. In some places you use the terms inter and intra rater and then in the conclusion section of the abstract you talk about the aggregate scores and the variation between SLPs, I wonder if you include the inter/intra rater terms in brackets there so it is clear what you are describing.
To improve the message of this paper I would streamline all the different terminology, explain inter and intra rater and stick to those terms throughout. Sometimes this is not very clear and I am trying to guess when you are talking about various terms.
Introduction : the way you describe the percent estimation and the VAS, I struggled to tell how they differed. My initial reading suggested these were exactly the same with the only difference being how they are written down. However your discussion of the differences your results indicate between these suggest that the VAS looks more globally at dysarthria and the percent estimations are more focussed on intelligibility. This difference is not clear in your introduction and could improve reader understanding if the difference between the two is made clearer if possible.
Clinical implications section: this reads in a very positive light when I'm not sure it should be quite so positive. My take home message is really that single SLP judgement between SLPs was variable and not reliable. This should be an important clinical message to say SLPs should consider their own assessment and judgement due to individual variation. This is crucial as clinical practice is almost always a single SLP so I can't see how aggregate scores are that applicable in clinical practice . However I think that the important clinical message is that a single SLP will be reliable in their own judgement (intra-rater) on both VAS and percent estimations, surely this has more important clinical implications as much more likely in clinical practice? I do think it raises for the reader but not really addressed or mentioned a very important point about what measures are important to the people we are working with in clinical practice and I sense this should be an important point in your conclusions.
I also wonder if in your limitations section you mention the 'objectiveness' or otherwise of the orthographic transcription. You describe this as the objective measure that the subjective SLP tasks were measured against but I think there may be more nuance around how truly objective it is.
I hope these comments can improve this paper and what it could mean for SLPs.
Reviewer 2 Report
Summary of article: This manuscript examined the reliability and validity of SLPs’ estimations of speech intelligibility in speakers with dysarthria secondary to Parkinson’s disease, Huntington’s disease, amyotrophic lateral sclerosis, and cerebellar ataxia. SLPs’ VAS ratings and estimations of intelligibility were compared to naïve listeners’ orthographic transcriptions. Intra and inter-rater reliability of the SLPs’ ratings and estimations were examined. Overall, SLP judgements were found to be both valid and reliable.
Strengths: This paper begins to answer a very important clinical question. The rationale for evaluating clinicians’ estimations of intelligibility, as it is the most widely used method for documenting intelligibility in the clinic, is excellent, as distinguished from estimations of intelligibility from VAS ratings (as a widely used method in research). The use of speakers with different types of dysarthria secondary to a variety of etiologies is a strength of this study, as is the sample size of the listener groups.
Weaknesses:
· Overall, the title, abstract, and introduction/rationale really focus on evaluating the reliability and validity of speech intelligibility estimations. But I feel like this gets lost in the research questions/methods and is not adequately connected to the discussion. For example, which of the methods (in the data analysis) are related to your question of validity and which type of validity is being addressed? How/what does determining the relationship between orthographic transcriptions and the VAS/estimations tell us about validity? And then in the discussion, what kind of evidence do these findings provide for validity?
o Somewhere (possibly in the introduction and/or discussion) I think you want to be clear about what you mean by validity (i.e., what type) and how this might be affected by different types of analyses (i.e., correlation analyses vs. exact agreement, etc.).
· I’m not sure I totally agree that (starting at line 111), SLPs’ ratings should ideally align with naïve listeners’ ratings. This may be the case when, as you describe, you want an idea of functional communicative ability in the “real world”, but maybe it’s okay that their ratings differ – do ratings from different types of listeners provide complementary information about communicative function? In my opinion, a more nuanced discussion of this is needed, rather than a statement saying that it’s ideal that SLP and naïve listener ratings should be similar, especially when SLPs are highly trained and thus, their ratings SHOULD be different than naïve listeners. I know this is a subtle distinction, but I think it’s an important one.
· The stimuli used for this study requires some additional thought/discussion. In particular, there a few concerns I have:
o What did the phrases of the passage look like and how might hearing one phrase over another impact ratings/judgements? Since we know parameters like stimuli length, predictiveness, lexicality, phonetic complexity, emotionality, etc. can impact intelligibility, could there have been an impact of differing stimuli on ratings?
o Most SLPs would know the Grandfather passage quite well; how might this impact the SLPs’ ratings and would this be different than the naïve listeners who likely are unfamiliar with the Grandfather passage?
o It seems to me that comparing the task for the naïve listeners to the task for the SLP listeners is not comparing oranges to oranges. For example, the naïve listeners had a lot more speech material (i.e., ½ the passage) than the SLPS (i/e., ½ passage). How might this impact your findings?
· In general, the figures and tables require more information to be interpretable on their own. For example, in Figure 2 what is the difference between the violin plot and the individual data points? Does the color gradation of the individual points mean anything? What does each data point refer to (i.e., one speaker vs. one listener)? What does the box represent (as well as the line in the box and the hinges and whiskers)? Also, a unit on the intelligibility (i.e., “%”) axis would be helpful. As another example, in Figure 4, it would be helpful to have in the note why HDM10 and AM1 are labeled on the graph. I know you talk about it in the text later, but I was confused when I first saw the figure.
o Table 2 displays findings from model 2, but where are the results from model 1?
o Figure 3 – I think it would be more helpful to see individual data points (i.e., in a scatter plot) than individual SLP correlations because this would allow the reader to see the actual spread of the data. Perhaps each SLP could have their data points in a different color if seeing the individual SLP data is of importance. As written, I don’t see the need for seeing individual SLP correlations. If the authors wish to keep this figure as is, a clearer rationale is needed.
· I appreciate the clinical implications section and the clear discussion about not replacing orthographic transcriptions with estimations of intelligibility. But what DOES the current work imply for clinical practice then? Since, as discussed in the introduction, SLPs aren’t using transcription clinically, what do the current findings suggest? In the introduction and discussion, the authors write that “before these subjective intelligibility measures can be adopted as a replacement for formal measures that utilize transcription…” (lines 136-137), but based on the work by King et al. (2012), it seems like transcription has already been replaced in the clinical world.
Minor concerns:
· Abstract:
o It would be nice to see that you included speakers with different etiologies of dysarthria; this is a strength of the paper and could be highlighted here
· Introduction:
o Lines 39-41 – if you want to make the case that lack of access to naïve listeners is a barrier to using formal intelligibility testing I think you need to make it clear that these tests require naïve listeners (or that naïve listeners would be ideal [vs. the treating clinician] to avoid familiarity effects). For example, an SLP could, in theory, use the SIT, but transcribe the sentences themselves.
o Line 29 – why not cite the newer version of the SIT from 2007?
o Lines 95-97 – or do you think it could be more about having a range of severity vs. different types of dysarthria?
o Lines 150-151 – citation needed; what ‘previous research’ is being referred to?
§ Same with Line 156
· Methods:
o If space is an issue, I think the racial/ethnic information of listeners could be removed. I would actually be more interested in the geographic location of the listeners (given that this could have an impact on dialects of English spoken).
o Line 195 – how was severity of dysarthria determined? I see that it was ‘based on baseline intelligibility testing’, but were there cutoffs for severity estimation? Also, I would like to see ranges and means of this baseline intelligibility testing to provide an indication of overall severity within each group. I know this can be inferred from Table 1, but how does ‘baseline intelligibility testing’ differ from the info in Table 1?
o Were listeners asked to wear headphones for the listening task? How many times could the listeners repeat each stimulus?
· Discussion:
o Lines 436-439 – this is interesting especially since the instructions for the VAS and the estimations were exactly the same. I wonder if, in the future, explicit discussions with SLPs about what they are rating.
· Limitations:
o Lines 499-500 – the point about recruiting a more diverse sample of SLPs in the future is well taken. But also consider that only using SLPS in medical settings may actually be more ecologically valid since these are the SLPs who are most likely seeing adult patients with dysarthria. If you had pediatric speakers, then looking at SLPs who work in schools might be more relevant. But I’m actually not sure how much value it would add to include SLPs in different settings.
· Conclusions:
o A clearer rationale for why investigating individual SLP ratings is needed (lines 518-522).
Reviewer 3 Report
- It is not quite sure why naive listeners were compared to SLP experts and why this information is clinically important.
-The Introduction is somehow missing the paragraph that would be devoted to the dysarthria test used in evaluating the dysarthria speech not just VAS and percent estimation intelligibility. And proper references. And maybe to draw some connection in the discussion section of the importance of the current research vs dysarthria tests (validated) used in the clinical evaluation.
-Figure 2 please better explain the sketch, and y-axis should be better presented to be more understandable. What the circled box mean, the vertical line in the box?
-The references in the manuscript text to “”Error!Reference source not found” is a bit defocusing the reading the manuscript. Can the authors write in different way?
-Figure 4, what HDM 10 and AM1 mean? Please explain it in the figure caption
-Figure 1-4 are not referenced in the manuscript text ?
- Tables are not referenced in the manuscript text?
-Row 90 check the font
-The discussion section, row 399-400 should be written more clearly and the references are missing.
-Row 401 Abur et al. should be referenced properly in the manuscript text according to the Brain sciences guidelines
-row 410-411 “. Our results are broadly consistent with this previous finding”. This sentence should be written more understandably and references are missing, somehow the authors are repeating (row 399-400) the same phrases.
- Row 466, 468 please check the referencing in the manuscript text how many authors are presented according to Brain sciences guidelines?
Round 2
Reviewer 1 Report
Dear Authors
Thanks for your responses to all of my comments. I can see you have now provided greater clarity related to my points. This paper reads much better and is a useful addition to the dysarthria literature.
Best wishes
Author Response
Thank you so much for your feedback. We are glad that we were able to incorporate your suggestions and comments to improve this manuscript.
Reviewer 2 Report
The authors are thanked for their careful revisions and thoughtful responses to the reviews. I believe the manuscript is much improved by the revisions. The addition of methodological details (i.e., explaining the modified version of the Grandfather passage, information about rating dysarthria severity, etc.) increases the replicability of the study. Table 1 is very helpful as well – for completeness, could validity also be included in this table? This may also assist with remediating my concern in the following paragraph. The supplemental tables and figures are also much appreciated and substantially improve understandability of the project. Revisions to the clinical implications are well-taken.
Thank you for clarifying the type of validity and how your research questions were created to assess convergent validity. I’m still missing the connection to validity in the discussion of this paper. In fact, if I’m correct, the term ‘validity’ does not appear anywhere in the discussion outside of the very first sentence. Given that validity appears to be a central issue in this manuscript, it should be discussed in this section of the manuscript relative to the results. Relatedly, the difference between validity and ‘accuracy’ (as in lines 550-552) is unclear – do the validity findings speak to this at all?
Minor concerns:
· Some changes in tense throughout (especially when talking about the current study; i.e., “The current study examines…” vs. “The current study examined…” ; “We hypothesize…” vs. “We hypothesized…” in which the latter case should be maintained throughout, as these types of activities have already been completed.
· A small note for supplementary table 1 – I think I can guess what the speaker codes mean, but it could be helpful to include a column for neurological diagnosis (or spell out what the codes mean in the note). I would also like to see the dysarthria severity determined by the two SLPs in this table.
· Lines 558-559 – “…indicating that these methods likely are possibly measuring…” awkward wording
Thank you for this excellent work that is an important contribution to our field.
Reviewer 3 Report
Authors responded to my concerns. Please check the overal manuscript after "accepting track changes". I did not find version without track changes (only highlighting changes) so it is a bit difficult to read the manuscript like this. For example check line 19 in abstract does the word "moderate" is starting the new sentence or?
Round 3
Reviewer 2 Report
Thank you for your careful revisions to my minor concerns from the previous version. I am highly satisfied with this final result and believe it is a strong contribution to our field.
Author Response
Thank you so much for your feedback and suggestions. We are glad that we were able to incorporate all your helpful comments to improve this manuscript. Thank you again.
Reviewer 3 Report
The authors responded to my concerns.
Author Response
Thank you for your helpful comments and suggestions. We are glad that we were able to address your concerns and incorporate your feedback to strengthen this manuscript. Thank you again.